# Metal Complexes of Redox Non-Innocent Ligand *N*,*N*′-Bis(3,5-di-*tert*butyl-2-hydroxy-phenyl)-1,2-phenylenediamine

**DOI:** 10.3390/molecules29051088

**Published:** 2024-02-29

**Authors:** Ari Lehtonen

**Affiliations:** Intelligent Materials Chemistry Research Group, Department of Chemistry, University of Turku, FI-20014 Turku, Finland; ari.lehtonen@utu.fi

**Keywords:** redox non-innocent ligand, transition metal, complexes, coordination compounds

## Abstract

Redox non-innocent ligands react with metal precursors to form complexes where the oxidation states of the ligand and thus the metal atom cannot be easily defined. A well-known example of such ligands is bis(*o*-aminophenol) *N*,*N*′-bis(3,5-di-*tert*butyl-2-hydroxy-phenyl)-1,2-phenylenediamine, previously developed by the Wieghardt group, which has a potentially tetradentate coordination mode and four distinct protonation states, whereas its electrochemical behavior allows for five distinct oxidation states. This rich redox chemistry, as well as the ability to coordinate to various transition metals, has been utilized in the syntheses of metal complexes with M_2_L, ML and ML_2_ stoichiometries, sometimes supported with other ligands. Different oxidation states of the ligand can adopt different coordination modes. For example, in the fully oxidized form, two N donors are sp^2^-hybridized, which makes the ligand planar, whereas in the fully reduced form, the sp^3^-hybridized N donors allow the formation of more flexible chelate structures. In general, the metal can be reduced during complexation, but redox processes of the isolated complexes typically occur on the ligand. Combination of this non-innocent ligand with redox-active transition metals may lead to complexes with interesting magnetic, electrochemical, photonic and catalytic properties.

## 1. Introduction

In 1966, Jørgensen introduced the term “non-innocent ligand” in coordination chemistry, referring to the uncertainty of the formal oxidation states of the central atom and the coordinated ligands [1]. By definition, a ligand is called “redox non-innocent” if it reacts with metal precursors to form complexes where the oxidation states of the ligand and therefore the central metal atom cannot be unquestionably defined [2,3]. Such redox non-innocent ligands can be, for example, simple molecules, e.g., O_2_, NO, and CO; synthetic unsaturated molecules, e.g., catechols or o-aminophenols; proteins; or other large biomolecules found in metal enzymes [4,5]. In metal complexes, redox processes typically occur at the metal center instead of at the classical “innocent” ligands. Conversely, redox non-innocent ligands have more available energy levels to participate in reduction and oxidation events; therefore, ligand-centered redox processes can also occur, with the oxidation state of the metal center remaining, or both the ligand and metal can change oxidation states. Metal complexes with non-innocent ligands have been studied, for example, as model compounds for metalloenzymes or as new homogenous catalysts, mainly since they can act as electron reservoirs in a catalytic cycle. The redox activity of these complexes permits redox processes and thus bond breaking and forming between the substrate and the central metal, which is essential for catalytic processes [6,7]. Complexes with redox non-innocent ligands may carry electrons with unpaired spins on the metal orbitals and also on the molecular orbitals of the organic ligands, which can cause the complexes to have interesting magnetic properties due to intramolecular (anti)ferromagnetic coupling [8,9].

As many redox non-innocent ligands are aromatic or other unsaturated molecules, it is possible to draw a number of resonance structures for such metal organic complexes. In practice, however, it is quite rare to see compounds that are truly isolable in all these theoretical resonance structures. As a representative example of aromatic redox non-innocent ligands, 2,4-di-*tert*-butyl-6-(phenylamino)phenol (H_2_NO^t^Bu) (Figure 1) and its derivatives have been a subject of interest owing to their stability and simple syntheses. Commonly, the proligand H_2_NO^t^Bu and its derivatives are synthesized by a reaction between 3,5-di-*tert*-butylcatechol and a primary aromatic amine, typically prior to complexation, but sometimes *in situ*. Although metal complexes with non-innocent ligands are generally used for catalytic applications [7,10], a number of complexes of H_2_NO^t^Bu and its derivatives have also been studied to find, for instance, new magnetic materials [11,12]. For example, H_2_NO^t^Bu is known to form a low-spin Co(III) complex, Co(NO^t^Bu)_3_ (**1**), where three identical ligands coordinate to the metal ion as bidentate iminosemiquinone radical anions and three iminosemiquinone radical ligands are ferromagnetically coupled [13].

Calculation of the formal oxidation states of metals and ligands is an important tool used in inorganic chemistry to understand and predict chemical reactivity. In particular, understanding the principal reactions in homogenous catalysis depends strongly on the valence electron count and the determination of oxidation states. By definition, the non-innocence of the ligand does not allow for a simple calculation of the oxidation states of the ligand and the metal center. However, it is possible to use some empirical approximations, such as metrical oxidation state (MOS) calculations, initially proposed by Brown in 2012, to evaluate the formal oxidation states of the metal-coordinated catechol and *o*-aminophenol moieties [14]. These estimates are based on the geometrical parameters obtained from high-quality single-crystal X-ray diffraction data, particularly the bond lengths of O-C and N-C bonds, and the C-C bonds of the phenyl rings [15].

*N*,*N*′-bis(3,5-di-*tert*-butyl-2-hydroxyphenyl)-1,2-phenylenediamine (Figure 2, H_4_N_2_O_2_, from here on N_2_O_2_ corresponds to the deprotonated ligand in any oxidation state) was first reported in 1999 by Wieghardt et al. It was originally synthesized to prepare Cu and Zn complexes, which can engage in homogenous catalysis where alcohols are oxidized by molecular oxygen in an ambient atmosphere [16]. H_4_N_2_O_2_ can be considered as a top of the *o*-aminophenol-based non-innocent ligands as it combines two *o*-aminophenol moieties and exists in five protonation states as well as at five different oxidation levels, which range from the oxidation state of 0 to −4 (Figure 3). Electron delocalization in the molecule may also span over ten atoms, from one phenolic oxygen atom to another. The oxidation state of the ligand strongly affects the coordination properties, as the donor atoms can be considered either as neutral or anionic; therefore, the metal-to-donor distances may vary remarkably. Moreover, in its partially or fully oxidized forms, one or two N donors are sp^2^-hybridized, which makes the ligand planar, whereas in the reduced form, sp^3^-hybridized N donors allow the formation of more flexible chelate structures. As a result, the proligand H_4_N_2_O_2_ has gained attention from coordination chemists over the years, and it has proven its effectiveness, for example, in homogenous catalysis [16,17,18,19]. The proligand is synthesized and isolated separately, rather than in situ, to form complexes directly. Nevertheless, the synthesis is simple; i.e., two equivalents of 3,5-di-*tert*-butylcatechol are mixed with one equivalent of *o*-phenylenediamine in a basic hydrocarbon solution under ambient conditions [16]. In addition to the redox processes and the formation of radicals, H_4_L can undergo other structural reorganizations upon coordination [20,21].

In addition to complexation studies, H_4_N_2_O_2_ has also been used as a precursor to form a heterocyclic carbene ligand which coordinates to metal ions as a tridentate ligand with two phenolic oxygens and a carbene moiety (Figure 1) [23]. Related transition metal complexes have been studied as catalysts in polymerization reactions [24,25] and reversible alkyl migration [23] within the complex.

Since the original publication by Wieghardt et al. on Cu and Zn complexes [16], a number of coordination compounds of H_4_N_2_O_2_ with other metals have been reported, namely Ti [17,26,27], Zr [17,27,28], Hf [17,26], V [27], Mo [29], W [30], Mn [20], Fe [20,21], Co [31], Ni [27], Sn [32] and U [33]. In addition, several compounds of P [22,34] and Te [35] are also known. Until now, reports on metal complexes with H_4_N_2_O_2_ have been focused on the structural and spectral characterization of the isolated products, but studies on chemical reactivity, including catalytic applications, are still rare. The most notable catalytic uses to date are the use of copper complexes for the aerobic oxidations of alcohols to aldehydes [16,19] and aliphatic hydrocarbons to alcohols [18], zirconium-catalyzed disproportionation of 1,2-diphenylhydrazine [28] and the polymerization of *rac*-lactide using titanium and zirconium complexes [17]. As H_4_N_2_O_2_ can form stable radicals upon coordination to the metal centers, such complexes might be useful in materials research. In general, stable neural radicals can offer building blocks to construct materials for magnetic, spintronic, electronic and optoelectronic applications [36].

## 2. Metal Complexes of *N*,*N*′-Bis(3,5-di-*tert*-butyl-2-hydroxyphenyl)-1,2-phenylenediamine

### 2.1. Group 4 Complexes: Ti, Zr, Hf

Roughly one half of all structurally characterized metal complexes of H_4_N_2_O_2_ are compounds of the group 4 transition metals Ti, Zr and Hf. They have been studied, for example, to find new catalysts for the polymerization of lactide to prepare biodegradable polymers derived from renewable resources [17], whereas another known catalytic application for such complexes is the two-electron disproportionation of diphenylhydrazine to azobenzene and aniline [28]. In the syntheses of group 4 metal complexes, the nature of the metal precursors seems to have a crucial role in the reaction outcome. For example, M(O^t^Bu)_4_ (M = Ti, Zr), which carry bulky alkoxide groups, react with H_4_N_2_O_2_ under inert conditions to give exclusively monomeric complexes with the formula of M(H_2_N_2_O_2_^Red^)(O^t^Bu)_2_ (M = Ti (**2**), Zr (**3**)). However, when less sterically hindered alkoxide Ti(O^i^Pr)_4_ is used under identical reaction conditions, two distinct complexes of different stoichiometries are formed depending on the reactant ratio. Specifically, a dinuclear complex (µ-H_2_N_2_O_2_^Red^)(Ti(µ-O^i^Pr)(O^i^Pr)_2_)_2_ (**4**) is formed with a 1:2 ligand-to-metal ratio, whereas the mononuclear Ti(H_2_N_2_O_2_^Red^)(O^i^Pr)_2_ (**5**) can be obtained using a stoichiometric amount of the metal alkoxide and ligand (Figure 2). Interestingly, these complexes can be easily converted to each other, i.e., adding one equivalent of Ti(O^i^Pr)_4_ to the solution of **5** yields the dinuclear complex **4**, whereas adding one equivalent of H_4_N_2_O_2_ to the dinuclear **4** yields the mononuclear complex **5**. The oxidation state of the metal center remains at +4, so there are no redox processes associated with these transformations [17]. Noteworthily, in both kinds of complexes, the H_2_N_2_O_2_ ligand is bonded in the reduced dianionic form; thus, these are rare examples where the N donors of the ligand are sp^3^-hybridized so they can be rather freely arranged around the metal centers. In further studies, the monomeric complexes were used as active catalysts for the polymerization of lactic acid, whereas the dinuclear complexes were less active [17].

In the solid state, the isostructural complexes M(H_2_N_2_O_2_^Red^)(OR)_2_ are made of molecular units with an octahedral *fac-fac* geometry, where phenolate oxygens, alkoxide oxygens and nitrogen donors are in *trans*, *cis* and *cis* positions, respectively. On the contrary, it was found that both metal centers in (µ-H_2_N_2_O_2_^Red^)(M(µ-OiPr)(O^i^Pr)_2_)_2_ have distorted octahedral coordination spheres of O_5_N donor sets, in which two phenol arms extend from the same face of the phenylenediamine ring. The bridging alkoxides enable the formation of a four-membered M-O-M-O ring with an M-M distance of ca. 3.3 Å (Figure 4).

The complexes M(N_2_O_2_^Red^)(L)_n_ (M = Ti (**7**), L = pyridine (py), n = 2; M = Zr (**8**), Hf (**9**), L = tetrahydrofuran (thf), n = 3) can be made from the H_4_N_2_O_2_ ligand and the metal precursors MCl_4_(L)_2_ (L = py, thf) in the presence of additional coordinating molecules if needed. In all these complexes, the tetradentate N_2_O_2_^Red^ anion coordinates in the equatorial position (Figure 3), while the formal oxidation state of the metal center is +4 [26]. As the N donors are sp^2^-hybridized, the ligand is essentially planar. In the case of Ti complex **7**, two pyridine molecules are bonded to the metal center to make a distorted octahedral coordination sphere, whereas larger Zr and Hf atoms in **8** and **9** can occupy three molecules of tetrahydrofuran to form heptacoordinated complexes, where two neutral ligands are coordinated in an axial position and one ligand is coordinated in an equatorial position to form a pentagonal-bipyramidal structure. All complexes can further react with a chlorine-based oxidant, iodobenzene dichloride PhICl_2_, to yield M(N_2_O_2_^Ox^)Cl_2_(L) (Ti (**10**), Zr (**11**), Hf (**12**)), where the non-innocent ligand is oxidized by two electrons (Figure 3) [26,28]. Interestingly, efforts to produce odd-electron species by the addition of 0.5 or 1.5 equivalents of PhICl_2_ only results in lower yields of the two-electron oxidized products, while there is no evidence of the one-electron oxidation products M[N_2_O_2_^Sq1·^]Cl(L)_2_. Zirconium complex **8** was found to catalyze a multielectron transfer reaction, wherein 1,2-diphenylhydrazine disproportionates to form aniline and azobenzene via a putative metal-imide intermediate. As the oxidation state of the metal does not change during the catalytic process, this was the first example of a d^0^ metal complex that can catalyze a multielectron reaction through the use of ligand-based valence changes [28]. Another interesting feature of compounds **7**–**9** is their strong NIR absorption around 930 nm with a high-energy shoulder near 750 nm, which results in the observed dark green color of the oxidized complexes. A similar absorption is not observed for the above-mentioned complexes with H_2_N_2_O_2_^Red^, which is clearly seen when yellow **8** is oxidized to yield dark green **11 [28]**.

M(N_2_O_2_^Ox^)_2_ (M = Ti (**13**), Zr (**14**)) can be made in basic methanol solutions of simple non-alkoxide metal precursors TiOSO_4_ and ZrOCl_2_, respectively (Figure 4) [27]. The mononuclear complexes are formed with a 2:1 ligand-to-metal ratio regardless of the equivalent ratios of the starting compounds. The C-N and C-C distances in the central phenylene rings of each distinct ligand suggest a localized, cyclohexadiene diimine-like structure rather than a delocalized system, which clearly indicates a partially oxidized ligand form. As the reactions were run in an ambient atmosphere, the oxidation of the ligand is caused by atmospheric oxygen instead of high-valent metal centers. In isostructural complexes, both ligands are coordinated to the metal center as fully deprotonated, nearly planar, tetradentate, dianionic ONNO donors. This results in eight-coordinated complexes, where the coordination geometry is best described as a distorted square antiprism and where the phenolate oxygens are positioned in a *mer* fashion (Figure 4).

In general, H_4_N_2_O_2_ seems to react with different group 4 metal precursors to form diamagnetic M(IV) species, whereas all the redox events upon coordination and further reactions occur on the ligands. Also, cyclic voltammetry indicates that the reversible electron transfer processes for the isolated complexes are mostly based on the ligands.

### 2.2. Group 5 and 6 Complexes: V, Mo, W

The only group 5 complex of the title ligand reported so far is V(N_2_O_2_^Sq1^)(HN_2_O_2_^Ox^) **15**, which can be made of either VO(acac)_2_ or VOSO_4_ and two equivalents of the ligand precursor in a basic methanol solution (Figure 5) [27]. As the synthesis can be achieved in an ambient atmosphere, partial oxidation of the ligand is achieved by atmospheric oxygen while the oxidation state of the metal ion remains the same. The complex is formed from neutral molecular species, where the two ligands show different coordination modes and protonation states; i.e., one of the two distinct ligands is coordinated to the metal as a fully deprotonated tetradentate ONNO donor, whereas the other ligand is coordinated as a partially deprotonated tridentate ONN donor and carries a dangling phenol part with an intact OH group. The phenolate oxygens are aligned in a *mer* fashion, whereas the geometry of the central V atom is best described as a distorted pentagonal bipyramid. The tridentate ligand has a planar geometry and the tetradentate one is moderately twisted. In the solid state, the dangling OH group forms an intramolecular interligand hydrogen bond to the phenolate oxygen of the four-dentate ligand. Compound **15** consists of an antiferromagnetically coupled radical ligand and the V^4+^ metal ion with one unpaired electron. It has a low paramagnetic signal in SQUID measurements due to a low-lying triplet state which is thermally populated [27]. Compound **15** has a broad absorption at λ_max_ = 680 nm that reaches the NIR range. The interaction with NIR radiation was further studied by measuring the magnetic properties under irradiation in the NIR region (λ = 785 nm), in which it was seen that the absorption of NIR photons leads to paired electrons. The measurements show that the magnetic moment, and therefore the number of unpaired spins, decreases within the molecule upon illumination. As this effect is not permanent at any temperature, the complex could be used to continuously convert IR quanta to paired electrons.

The group 6 metal complexes Mo(N_2_O_2_^Red^)(HN_2_O_2_^Sq1^) **16** [29] and W(N_2_O_2_^Red^)(HN_2_O_2_^Sq1^) **17** [30] are isostructural with **5**, even if the oxidation states of the ligands are different. The Mo complex is formed by the reaction of MoO_2_(acac)_2_ or MoO_2_(Heg)_2_ (Heg^−^ = ethanediolate monoanion) with two molecules of the ligand precursor in methanol, while the W complex is made of tungsten(VI) trisglycolate W(eg)_3_ (Figure 6). MOS calculations indicate Mo(VI) and W(VI) species, so both compounds formally have a d^0^ metal center. However, they both exhibit a paramagnetic response in SQUID, which is associated with one unpaired electron. The X-band EPR spectra for Mo and W compounds show signals at (S = 1/2) g_iso_ = 2.0087 and g_iso_ = 1.986, respectively, with minor asymmetry. These results support the idea that the radicals are ligand-based, though the low experimental g value for the W compound indicates some spin density on the central metal atom. The other known Mo(VI) compound with the title ligand is diamagnetic Mo(N_2_O_2_^Red^)Cl_2_(dmf) **18** (dmf = dimethyl formamide), which is made using MoO_2_Cl_2_(dmf)_2_ as a metal source [29]. In this compound, the seven-coordinated Mo center shares a plane with two oxygens and two nitrogens from the N_2_O_2_ ligand as well as with an oxygen donor from the coordinated dmf. Two chlorides in axial positions complete the distorted pentagonal bipyramidal coordination sphere around the metal center (Figure 5).

### 2.3. Group 7–10 Complexes: Mn, Fe, Co, Ni

The first-row metals Mn, Fe, Co and Ni form isostructural complexes M(HN_2_O_2_)_2_, although the ligand oxidations states vary depending on the metal [20,21,27,31]. All complexes are made by the simple reactions of metals salts with H_4_N_2_O_2_ in MeOH (Figure 7). In the solid-state structures, four N and two O donor atoms from two triply deprotonated tridentate ligands occupy the distorted octahedral coordination sphere around the metal center. Deprotonated phenolate O atoms are coordinated in a *cis* arrangement, resulting in an approximately C_2_ symmetric complex. One phenolic OH from each of the ligands remains protonated and forms an intramolecular interligand hydrogen bond.

The Mn complex **19** is only briefly mentioned in one report, and no detailed discussions on the structure or the metal and ligand oxidation states were included [20]. However, the MOS calculations based on the geometrical parameters extracted from the single-crystal XRD data indicate an oxidation state of +III for the metal center, identical to the isostructural low-spin Fe(III) (**20**) and Co(III) (**21**) complexes [21,31]. In all these Mn, Fe and Co complexes, the ligands exhibit a HN_2_O_2_^ox^/HN_2_O_2_^sq1^ resonance structure; i.e., one ligand is oxidized by two electrons and the other ligand by one electron. Contradictorily, in the isostructural Ni complex, the two ligands are identical (Figure 6, Figure 8). The shortest C-N bonds are towards the central phenylene ring of the ligand, corresponding to imine double bonds, so the Ni complex consists of one Ni^2+^ ion and two HN_2_O_2_^Ox^ ligands. Actually, Ni(HN_2_O_2_^Ox^)_2_ (**24**) is isostructural and isomorphous with **22**, which clearly demonstrates that the overall structure and geometry of the complex are not dependent on the oxidation level of the ligands.

Trivalent Mn, Fe and Co compounds are made under ambient conditions using MnCl_2_·4H_2_O, anhydrous FeCl_3_ and CoCl_2_·6H_2_O, respectively, so the ligand and metal ions can again be oxidized by atmospheric oxygen upon complexation. Interestingly, the redox-active ligand can also be “chemically non-innocent” and react further when coordinated. Precisely, when the reactions of Fe and Co salts with the ligand precursor were run in acetonitrile in the presence of triethyl amine, the coordinated ligands underwent a structural reorganization involving irreversible intramolecular cyclization to produce mixed phenolate/phenoxazinyl radical species (Figure 7) [20,21]. To our knowledge, there are no more reports on intramolecular reactions with any other complexes of H_4_N_2_O_2_.

The Evans NMR method was conducted with **24** in a CDCl_3_ solution to measure µ_eff_ = 2.84 µ_B_ at room temperature. The solid-state SQUID measurements gave essentially the same value, µ_eff_ = 2.88 µ_B_, which is in a good agreement with an octahedral high-spin Ni^2+^ moiety, so **24** is a paramagnetic, high-spin Ni(II) complex. It has a strong near-IR absorption at 830 nm. When the magnetic moment was measured under NIR radiation (λ = 785 nm), it was found to increase; therefore, the Ni complex seems to convert photons to free electrons [27].

There is a brief comment on one more Fe complex of H_4_N_2_O_2_, namely Fe(N_2_O_2_)Cl (**25**). The overall structure resembles a mononuclear molecule with a distorted square pyramidal geometry around the pentacoordinated central atom, but so far there are no structural data available for further discussion [20].

### 2.4. Group 11,12,14 Complexes: Cu, Zn, Sn

In their seminal paper, Wieghardt et al. described the Cu and Zn complexes in four out of five theoretically possible different oxidation states of H_4_N_2_O_2_ [16]. These compounds were studied as homogenous catalysts for the atmospheric oxidation of primary alcohols with concomitant formation of H_2_O_2_ and aldehydes. These four oxidation states of the ligand were obtained using various counterions and different reaction conditions, although most of the products were sensitive to air and were not stable enough for a full structural characterization. The stable molecular M(II) complexes, paramagnetic Cu(N_2_O_2_^Ox^) **26** and diamagnetic Zn(N_2_O_2_^Ox^) **27**, were isolated and fully characterized (Figure 9). The isostructural complexes carry a ligand containing a central diiminoquinone moiety and two phenolates, so they are in a fully deprotonated and doubly oxidized form (Figure 7). The (N_2_O_2_^Ox^)^2−^ ligand coordinates in a tetradentate fashion, whereas the MN_2_O_2_ coordination polyhedron is nearly square planar. The cyclic voltammograms of **26** and **27** are very similar, showing two one-electron reduction and two one-electron oxidation waves, which indicate ligand-centered redox processes. Reaction of the neutral complexes M(N_2_O_2_^Ox^) in CH_2_Cl_2_ with ferrocenium hexafluorophosphate yields M(N_2_O_2_^Sq2^)PF_6_. NMR studies on the Zn(II) complexes also indicate the formation of a neutral species Zn(H_2_N_2_O_2_^Red^) (**28**), although it could not be isolated. The anionic complexes [M(N_2_O_2_^Sq2^)]PF_6_ (M = Cu (**29**), Zn (**30**)) were formed in the solutions. They were found to be capable of oxidizing primary alcohols to the corresponding aldehyde while forming the two-electron reduced species [M(N_2_O_2_^Sq1^)]^−^, where the ligand is a paramagnetic organic radical. Magnetic susceptibility measurements show a temperature-independent magnetic moment of 1.8 µ_B_ for **30**. For solid **29**, the measurements show the temperature dependence of the effective magnetic moment, which arises from the intermolecular antiferromagnetic coupling between the organic radical and the Cu^2+^ ion. Compound **26** was later studied as a biomimetic galactose oxidase model compound and was found to be capable of aerobically oxidizing the *iso*-butane C-sp^3^-H bond to yield *tert*-butanol with a high selectivity [18]. Moreover, the property of **26** to oxidase alcohols was further explored to use it as a catalyst in a flow microreactor to oxidize benzyl alcohol and related substrates to corresponding aldehydes with high yields and selectivities [19].

The diamagnetic complex Sn(N_2_O_2_^Ox^)_2_ (**31**) (Figure 7) was obtained by the reaction of two equivalents of H_4_N_2_O_2_ with SnCl_4_ in the presence of triethylamine in toluene. The strongly colored compound was made in a two-step reaction, where the final product was obtained after the oxidation of an unstable intermediate by an equivalent amount of O_2_ [32]. In the isolated solid, the coordination sphere around the metal center is a distorted trigonal dodecahedron, instead of distorted square antiprism as found for the related Ti and Zr complexes [27]. Cyclic voltammetry analysis shows an irreversible two-electron process, which leads to the decomposition of the complex into a mixture of products. The chemical oxidation of **31** with AgBF_4_ in THF generates a paramagnetic cation, which was characterized by EPR spectroscopy to be [Sn(N_2_O_2_^Ox^)_2_]^+^, although the interpretation is not clear and [Sn(N_2_O_2_^Ox^)(N_2_O_2_^Sq2^)]^+^ is also proposed as a possible structure.

### 2.5. Actinoids: U

The diamagnetic uranium(VI) complex UO_2_(N_2_O_2_^Ox^)(dmso) (**32**) (dmso = dimethyl sulphoxide) was made from UO_2_(ClO_4_)_2_·nH_2_O and the ligand precursor in an MeCN solution followed by crystallization from a dmso/water mixture. [33] The crystalline product was not suitable for a closer examination of the bonding parameters, but re-crystallization from a pyridine/water mixture gave good-quality UO_2_(N_2_O_2_^Ox^)(H_2_O)·0.5C_5_H_5_N (**33**) crystals. The central U^6+^ ion has a pentagonal bipyramidal coordination geometry with a water molecule and the ligand ONNO donor in equatorial positions (Figure 8). The UV-Vis spectrum of **32** shows an absorption band at ca. 440 nm (ε = 10,000 M^−1^cm^−1^) due to the ligand-based π radical. Cyclic voltammetry and spectroelectrochemical studies indicate that UO_2_(N_2_O_2_^Ox^)(dmso) can be reversibly reduced by one or two electrons without causing any f–f transition that is visible in the UV-Vis spectra, so the electrochemical events occur solely on the non-innocent ligand while the oxidation state of U remains unchanged [33].

### 2.6. Metalloids: P, Te

The phosphorus compound P(N_2_O_2_^Red^)H (**34**) can be made from the reaction of H_4_N_2_O_2_ with PCl_3_ in tetrahydrofuran at room temperature using Et_3_N or *N*,*N*-diisopropylethylamine as a base; two research groups published its synthesis and full characterization at the same time independently [22,34]. Deprotonation with potassium bis(trimethylsilyl)amide gives a tricoordinated P(III) species P(KN_2_O_2_^Red^) (**35**), which can be further converted to K(18-crown-6)[P(N_2_O_2_^Red^)] (**36**) by adding crown ether to the reaction mixture (Figure 9, Figure 10) [34]. Alternatively, the reaction of **34** with N-chlorosuccinimide gives P(N_2_O_2_^Red^)Cl (**37**), which reacts further to yield P(N_2_O_2_^Red^)(CN) (**38**) [22]. Compound **34** and its pentacoordinated substitution products have a distorted square pyramidal geometry around the central atom. The ligand is unquestionably in a reduced form, and thus a tetra-anion, but the oxidation state of the phosphorus ion is debatable. In detail, Volodarsky et al. described **34** as a P(III) species [34], whereas Alcarazo et al. assigned it as a P(V) compound [22]. If the P-H hydrogen is considered as a hydride, the formal oxidation state is P(V). However, the reactivity of the compound described above (for instance, the hydrogen in **34** can be removed by a base or replaced by chlorine) supports the description as a P(III) compound. As ^31^P NMR chemical shifts, for example, are independent of formal oxidation numbers, it is hard to find spectroscopic evidence of the oxidation states of the central phosphorus atoms. Nevertheless, as these syntheses were run under inert conditions, the formal oxidation state of the P center is due to the oxidative addition of a NH or OH bond to the central atom [37].

The reaction of H_4_N_2_O_2_, TeCl_4_ and an excess of Et_3_N in tetrahydrofuran under air- and moisture-free conditions affords diamagnetic compound Te(N_2_O_2_^Ox^) (**39**), where the N_2_O_2_ ligand adopts an almost planar trapezoid-like arrangement around the central Te(II) ion (Figure 8) [35]. The complexation is associated with an intramolecular redox process, as the formal oxidation state of the Te ion is reduced by two electrons while the ligand is oxidized accordingly. The compound can be reduced by cobaltocene (Cp_2_Co) to obtain a paramagnetic salt [Cp_2_Co][Te(N_2_O_2_)] (**40**), where the central ion is Te(II) and the organic ligand is best described as a structural isomer of the [N_2_O_2_^Sq1^]^3−^ anion, so the redox process occurs again on the ligand [35].

## 3. Conclusions

Bis(*o*-aminophenol) *N*,*N*′-bis(3,5-di-*tert*butyl-2-hydroxy-phenyl)-1,2-phenylenediamine (H_4_N_2_O_2_) is a well-known redox non-innocent ligand, which has a potentially tetradentate coordination mode and five different protonation states, and its electrochemical behavior allows five separated oxidation states. As H_4_N_2_O_2_ has a rich redox chemistry and can form coordination compounds with different metals, it has been used in the syntheses of a number of metal compounds with M_2_L, ML and ML_2_ stoichiometries, sometimes supported with other ligands. Combination of H_4_N_2_O_2_ with redox-active transition metals leads to complexes with interesting magnetic, electrochemical, photonic and catalytic properties. The oxidation states of the ligands and metal centers seem to be dependent on the reaction conditions. Specifically, if reactions are run in an ambient atmosphere, atmospheric oxygen may participate in the oxidation of the metal or the ligand, leading to the formation of the final products. In general, the redox processes of the isolated compounds occur in the ligand part, whereas the oxidation states of the metals stay the same. So far, the applications of metal complexes with H_4_N_2_O_2_ are restricted to catalytic applications such as lactide polymerization and oxidation of alkanes or alcohols. However, the possibility to synthesize stable complexes which contain unpaired electrons in the ligand and/or metal orbitals may offer applications, for example, in single-molecule magnets and photonic materials.

## Data Availability

No new data were created.

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
