# Peer review of "Metal Complexes of Redox Non-Innocent Ligand *N*,*N*′-Bis(3,5-di-*tert*butyl-2-hydroxy-phenyl)-1,2-phenylenediamine"

_molecules, 2024, doi:10.3390/molecules29051088_

Round 1
Reviewer 1 Report
Comments and Suggestions for Authors
The review written by Ari Lehtonen report the progress of metal complexes of redox non-innocent ligand N,N′-bis(3,5-di-2-tertbutyl-2-hydroxy-phenyl)-1,2-phenylenediamine. There have been many scientific studies regarding the applications, mainly in magnetic, electrochemical, photonic and catalytic property, therefore a review accentuating the employment of the named ligands and their complexes is demanded. Although this review looks solid, there are still many problems that need to be solved. I would like not to recommend its publication in Molecules, a leading international journal of chemistry.
Some detailed comments are listed below:
1. The paper is written in a descriptive way, which makes it a bit difficult to understand. The writing style should be improved for better reading of this paper. As a review, more discussion, comparison and analysis may be required.
2. As a review, the application of these complexes should be added. The simple introductions for title-related compounds are not enough.
3. As a review, in order to better present the current research, a summary of some relevant studies may be required, including the Figures and Tables, especially the syntheses and structures.
4. What are the unique advantages of this ligand, which should be emphasized in the introduction.
5. For isomorphism, whether do different metals have different properties? In generally, what is the order? What is reason? I strongly encourage authors to reread the literature and summary it. What is the discovery and conclusion? In addition, some important Figures are needed, especially the representative work. Be aware of the copyright issues! It is recommended to reorganize it.
6. The current introduction is too simple. What are the main situations and challenges? What are the key scientific issues that need to be addressed? It is recommended to reorganize it.
7. I strongly encourage authors to find a native speaker for proofreading. There are many grammar, spelling and format mistakes in the current manuscript.
8. For the compounds exemplified in Part 2, specific structural diagrams and tables should be given, including structural information and outstanding properties of the compounds for easy understanding.
9. There are few references and further reading is needed.
10. Some texts in the “abstract” and “conclusion” sections are similar. A check is needed. In addition, the significance of this work should be sublimed in the “conclusion” section.
Comments on the Quality of English LanguageI strongly encourage authors to find a native speaker for proofreading. There are many grammar, spelling and format mistakes in the current manuscript.
Author Response
The review written by Ari Lehtonen report the progress of metal complexes of redox non-innocent ligand N,N′-bis(3,5-di-2-tertbutyl-2-hydroxy-phenyl)-1,2-phenylenediamine. There have been many scientific studies regarding the applications, mainly in magnetic, electrochemical, photonic and catalytic property, therefore a review accentuating the employment of the named ligands and their complexes is demanded. Although this review looks solid, there are still many problems that need to be solved. I would like not to recommend its publication in Molecules, a leading international journal of chemistry.
Reply: Thank you for the comments. The detailed replies are given above.
Some detailed comments are listed below:
- The paper is written in a descriptive way, which makes it a bit difficult to understand. The writing style should be improved for better reading of this paper. As a review, more discussion, comparison and analysis may be required.
Reply: Some text is added whenever relevant material is available.
- As a review, the application of these complexes should be added. The simple introductions for title-related compounds are not enough.
Reply: The applications are reviewed more thoroughly, but in fact there is not so much results to be discussed here.
- As a review, in order to better present the current research, a summary of some relevant studies may be required, including the Figures and Tables, especially the syntheses and structures.
Reply: It is true that in a review, more details on the current research could be added. However, as there are no specific suggestions it is rather hard to realise what kind summary the reviewer is thinking about. Anyway, more material is added per reviewer’s suggestions.
- What are the unique advantages of this ligand, which should be emphasized in the introduction.
Reply: This ligand does not necessarily have any unique advances, but is treated here as an “extreme” example on the redox non-innocent o-aminophenol and therefor worth of investigation.
- For isomorphism, whether do different metals have different properties? In generally, what is the order? What is reason? I strongly encourage authors to reread the literature and summary it. What is the discovery and conclusion? In addition, some important Figures are needed, especially the representative work. Be aware of the copyright issues! It is recommended to reorganize it.
Reply: I am not sure, what the reviewer is meaning. The properties of isomorphous compounds of different metals are discussed.
- The current introduction is too simple. What are the main situations and challenges? What are the key scientific issues that need to be addressed? It is recommended to reorganize it.
Reply: The introduction part is partially re-written and re-organised.
- I strongly encourage authors to find a native speaker for proofreading. There are many grammar, spelling and format mistakes in the current manuscript.
Reply: The language have been polishded.
- For the compounds exemplified in Part 2, specific structural diagrams and tables should be given, including structural information and outstanding properties of the compounds for easy understanding.
Reply: Unfortunately, I can not follow the idea of this comment.
- There are few references and further reading is needed.
Reply: Some general references on redox non-innocence as well as three more references on the title compound were added. Actually, practically all reliable literature on the use of title ligand is used.
- Some texts in the “abstract” and “conclusion” sections are similar. A check is needed. In addition, the significance of this work should be sublimed in the “conclusion” section.
Reply: The conclusion part is re-written and some more discussion is added.
Comments on the Quality of English Language
I strongly encourage authors to find a native speaker for proofreading. There are many grammar, spelling and format mistakes in the current manuscript.
Reply: The language has been polished.
Reviewer 2 Report
Comments and Suggestions for Authors
This manuscript from A. Lehtonen reviews the coordination chemistry and the redox properties of a particular class of phenylenediamine ligand, more specifically N,N’-bis(3,5-di-tBu-2-OH-phenyl)-1,2-phenylenediamine, prepared for the first time in 1999. The paper is almost comprehensive (see below) and covers adequately the timeframe until the present day. The ligand is of high interest, due to its versatility adopting different charges (able to behave as an electron reservoir) and using different donor atoms. It is this combination of the oxidation states and the coordination numbers which makes useful the ligand. The bonding behaviour of the ligand towards different metals and even representative elements has been adequately covered, although some improvement is still possible. Due to these facts, I recommend publication after revision. Therefore, the following aspects need to be considered.
1) When dealing with a single ligand within a very tight timeframe, one would expect the review to be comprehensive, and it is almost so. However, I believe that significant contributions have been omitted from the article. Perhaps the most important ones are: a) Dalton Trans. 2012, 41, 10970 (Pb); b) Appl. Organomet. Chem. 2021, 35:e6227 (Ti); c) Dalton Trans. 2021, 50, 6088 (V, it is from the author himself, why is it excluded?); d) Chem. Sci. 2022, 13, 9560 (catalysis); e) Ind. Eng. Chem. Res. 2022, 61, 13408 (also catalysis).
2) General presentation aspects: a) the described complexes should be numbered, as it would facilitate their tracking throughout the text; b) the X-ray structures presented should have labeled atoms, at least the most important ones, as in their current state, they are not informative at all.
3) Figure 3 is crucial as it helps to understand how many charges each N2O2 ligand (i.e., completely deprotonated) saturates while acting as a tetradentate ligand in different oxidation states. Perhaps it would be desirable to have an equivalent figure for partially deprotonated ligands H2N2O2 (where do the H atoms remain?) and HN2O2. I believe these figures are truly critical for understanding of the bonding mode adopted in each case.
4) In the schemes, in general, there are two syntheses; it would be convenient to distinguish between the right part and the left part. For example, figure 5 left, figure 5 right; scheme 5 left, scheme 5 right. Line 280 on page 9, it is figure 7. Line 287, cationic complexes (not anionic). Line 298, it is figure 7, not 6. Line 331, it is figure 9, not 8, the same in line 350.
Author Response
This manuscript from A. Lehtonen reviews the coordination chemistry and the redox properties of a particular class of phenylenediamine ligand, more specifically N,N’-bis(3,5-di-tBu-2-OH-phenyl)-1,2-phenylenediamine, prepared for the first time in 1999. The paper is almost comprehensive (see below) and covers adequately the timeframe until the present day. The ligand is of high interest, due to its versatility adopting different charges (able to behave as an electron reservoir) and using different donor atoms. It is this combination of the oxidation states and the coordination numbers which makes useful the ligand. The bonding behaviour of the ligand towards different metals and even representative elements has been adequately covered, although some improvement is still possible. Due to these facts, I recommend publication after revision. Therefore, the following aspects need to be considered.
1) When dealing with a single ligand within a very tight timeframe, one would expect the review to be comprehensive, and it is almost so. However, I believe that significant contributions have been omitted from the article. Perhaps the most important ones are: a) Dalton Trans. 2012, 41, 10970 (Pb); b) Appl. Organomet. Chem. 2021, 35:e6227 (Ti); c) Dalton Trans. 2021, 50, 6088 (V, it is from the author himself, why is it excluded?); d) Chem. Sci. 2022, 13, 9560 (catalysis); e) Ind. Eng. Chem. Res. 2022, 61, 13408 (also catalysis).
Reply: Thank you for these suggestions. We have tried our best to find all literature on the topic, but a few references are obviously missing. However, Dalton Trans. 2012, 41, 10970 reports tin and lead complexes with quite similar but still a different ligand, so it was not included in the review. Appl. Organomet. Chem. 2021, 35:e6227 seems to give one more example on the Ti complexes, but it is poorly characterised (e.g. MS and elemental analyses are erroneous), so it was also ignored. Chem. Sci. 2022, 13, 9560 (catalysis) and Ind. Eng. Chem. Res. 2022, 61, 13408 are reliable references and are now involved. In our own paper, Dalton Trans. 2021, 50, 6088, the V complex (which is originally reported in our other paper) is in a minor role and it was omitted to avoid unnecessary self-citations.
2) General presentation aspects: a) the described complexes should be numbered, as it would facilitate their tracking throughout the text; b) the X-ray structures presented should have labeled atoms, at least the most important ones, as in their current state, they are not informative at all.
Reply: The complexes are numbered and essential atoms in X-ray plots are labelled as suggested by the reviewer.
3) Figure 3 is crucial as it helps to understand how many charges each N2O2 ligand (i.e., completely deprotonated) saturates while acting as a tetradentate ligand in different oxidation states. Perhaps it would be desirable to have an equivalent figure for partially deprotonated ligands H2N2O2 (where do the H atoms remain?) and HN2O2. I believe these figures are truly critical for understanding of the bonding mode adopted in each case.
Reply: The figure was re-drawn to add protonic H-atoms, so it is easier to see where these atoms are placed in the oxidised molecules.
4) In the schemes, in general, there are two syntheses; it would be convenient to distinguish between the right part and the left part. For example, figure 5 left, figure 5 right; scheme 5 left, scheme 5 right. Line 280 on page 9, it is figure 7. Line 287, cationic complexes (not anionic). Line 298, it is figure 7, not 6. Line 331, it is figure 9, not 8, the same in line 350.
Reply: The scheme titles are completed.
Reviewer 3 Report
Comments and Suggestions for Authors
The presented manuscript reviews the chemistry of the ligand N,N'-bis(3,5-di-tert-butyl-2-hydroxyphenyl)-1,2-phenylenediamine with some metals of groups 4-12 and group 14 of the periodic table. Overall, I find this manuscript to be well-written showing the authors to deliver a comprehensive understanding of the topic. Nevertheless, I think that the introduction could benefit from a bit more inclusion of general literature references on the broader subject of this kind of ligands and I suggest the author to include them.
Moreover, the comments above should be addressed:
Line 123: The formation of a four-membered M-O-M-O ring cannot be seen in figure 3, as referenced.
Line 138: "while there is no evidence on one-electron oxidation products M[N2O2Sq1·]Cl(L)2" This sentence does not match with Scheme 2, as referenced.
Line 270: There are not reported synthesis of compounds of the ligand with metals of the group 13, so the title of the section 2.4 should be changed.
The used abbreviations of Dalton Transactions and Angewandte Chemie International Edition are incorrect in the bibliography.
Author Response
The presented manuscript reviews the chemistry of the ligand N,N'-bis(3,5-di-tert-butyl-2-hydroxyphenyl)-1,2-phenylenediamine with some metals of groups 4-12 and group 14 of the periodic table. Overall, I find this manuscript to be well-written showing the authors to deliver a comprehensive understanding of the topic. Nevertheless, I think that the introduction could benefit from a bit more inclusion of general literature references on the broader subject of this kind of ligands and I suggest the author to include them.
Reply: Some general literature was added and some more detailed introduction was written.
Moreover, the comments above should be addressed:
Line 123: The formation of a four-membered M-O-M-O ring cannot be seen in figure 3, as referenced.
Line 138: "while there is no evidence on one-electron oxidation products M[N2O2Sq1·]Cl(L)2" This sentence does not match with Scheme 2, as referenced.
Line 270: There are not reported synthesis of compounds of the ligand with metals of the group 13, so the title of the section 2.4 should be changed.
The used abbreviations of Dalton Transactions and Angewandte Chemie International Edition are incorrect in the bibliography.
Reply: Thank you, the mistakes pointed above are now corrected.
Reviewer 4 Report
Comments and Suggestions for Authors
The manuscript ”Metal complexes of redox non-innocent ligand N,N′-bis(3,5-di- 2 tertbutyl-2-hydroxy-phenyl)-1,2-phenylenediamine“ by Ari Lehtonen describes ligands that react with metal precursors to form complexes in which the degree of oxidation of the ligand and the metal atom is not easily defined.
I think the manuscript is interesting, but the author needs to complete the manuscript with publications from 2020-2023. The literature review should be based on the latest scientific research and include more literature.
It will also be useful to add to the manuscript a description of the use of the complexes from each group. Do the complexes only have applications in catalysis? What kind of processes are these catalysts used for: industrial or biological?
Author Response
The manuscript ”Metal complexes of redox non-innocent ligand N,N′-bis(3,5-di- 2 tertbutyl-2-hydroxy-phenyl)-1,2-phenylenediamine“ by Ari Lehtonen describes ligands that react with metal precursors to form complexes in which the degree of oxidation of the ligand and the metal atom is not easily defined.
I think the manuscript is interesting, but the author needs to complete the manuscript with publications from 2020-2023. The literature review should be based on the latest scientific research and include more literature.
Reply: Some new references are added per suggestions of reviewer. However, there is no more literature available – or it is very hard to find – on metal complexes of the title ligand.
It will also be useful to add to the manuscript a description of the use of the complexes from each group. Do the complexes only have applications in catalysis? What kind of processes are these catalysts used for: industrial or biological?
Reply: More discussion on the applications was added. In fact, these compounds do not have, so far, too many applications.
Round 2
Reviewer 1 Report
Comments and Suggestions for Authors
The current version may be acceptable.
Comments on the Quality of English LanguageThe current version may be acceptable.
Reviewer 4 Report
Comments and Suggestions for Authors
Accept in present form.